# Transfusion-Transmitted Malaria of *Plasmodium malariae* in Palermo, Sicily

**DOI:** 10.3390/healthcare9111558

**Published:** 2021-11-16

**Authors:** Jessica Pulvirenti, Maurizio Musso, Teresa Fasciana, Antonio Cascio, Maria Rita Tricoli, Natascia Oliveri, Maria Favarò, Orazia Diquattro, Anna Giammanco

**Affiliations:** 1A.O.U.P., Unit of Microbiology, Virology and Parasitology, 90127 Palermo, Italy; mariaritatricoli@gmail.com (M.R.T.); oliverinatascia@libero.it (N.O.); mariafavaro@libero.it (M.F.); 2UOC di Oncoematologia e TMO, Dipartimento Oncologico “la Maddalena”, 90146 Palermo, Italy; mamusso@libero.it; 3Department of Health Promotion, Mother and Child Care, Internal Medicine and Medical Specialties, University of Palermo, 90127 Palermo, Italy; teresa.fasciana@virgilio.it (T.F.); antonio.cascio03@unipa.it (A.C.); anna.giammanco@unipa.it (A.G.); 4Laboratory of Microbiology, Azienda Ospedaliera Ospedali Riuniti “Villa Sofia-V. Cervello”, 90146 Palermo, Italy; orazia.diquattro@villasofia.it

**Keywords:** malaria, blood transfusion, blood donor screening, *Plasmodium malariae*, transfusion-transmitted malaria, thrombocytopenia, asymptomatic semi-immune donors

## Abstract

Transfusion-transmitted malaria (TTM) is a rare occurrence with serious consequences for the recipient. In non-endemic areas, the incidence of transmission of malaria by transfusion is very low. We report a clinical case of transfusion-transmitted malaria due to *Plasmodium malariae*, which happened in a patient with acute hemorrhagic gastropathy. Case presentation: In April 2019, a 70-year-old Italian man with recurrent spiking fever for four days was diagnosed with a *P. malariae* infection, as confirmed using microscopy and real-time PCR. The patient had never been abroad, but about two months before, he had received a red blood cell transfusion for anemia. Regarding the donor, we revealed that they were a missionary priest who often went to tropical regions. *Plasmodium* spp. PCR was also used on donor blood to confirm the causal link. Discussion and Conclusions: The donations of asymptomatic blood donors who are predominantly “semi-immune” with very low parasitic loads are an issue. The main problem is related to transfusion-transmitted malaria. Our case suggests that *P. malariae* infections in semi-immune asymptomatic donors are a threat to transfusion safety. Currently, microscopy is considered the gold standard for the diagnosis of malaria but has limited sensitivity to detect low levels of parasitemia. Screening using serological tests and molecular tests, combined with the donor’s questionnaire, should be used to reduce the cases of TTM.

## 1. Introduction

Malaria is an infectious disease that is caused by intracellular protozoan parasites of the genus *Plasmodium* spp.; it is responsible for an acute febrile illness with varying severities depending on the species involved and on the subject’s immune status, but also on the process of invasion and multiplication of the parasite in human red blood cells (RBCs) during their complex life cycle. Five species of *Plasmodium* spp. are currently known to cause malaria in humans: *Plasmodium falciparum*, *Plasmodium vivax*, *Plasmodium malariae*, *Plasmodium ovale*, and *Plasmodium knowlesi*. Malaria is found in tropical and subtropical areas where malaria parasites are naturally transmitted by the infective bites of anopheles female mosquitoes during their blood meal. In many temperate areas, such as Western Europe and the United States, economic development and public health measures have succeeded in eliminating malaria. However, most of these areas have anopheles’ mosquitoes that can transmit malaria and the reintroduction of the disease is a constant risk [1]. Just as due to the presence of lice in the territory, new cases of recurrent fever transmitted by louse (LBRF) in Italy have been reported [2,3].

An alternative cause of malaria is transfusion-transmitted malaria (TTM). The incidence of transfusion-transmitted malaria in non-endemic nations due to severe donor choice is very low [4,5]. *Plasmodium falciparum*, *Plasmodium vivax*, and *Plasmodium malariae* are the species that are most frequently detected in TTM [6]. We present a case of malaria that was caused by *P. malariae* associated with transfusion in a patient after having acute hemorrhagic erosive gastropathy.

## 2. Case Presentation

In April 2019, a 70-year-old male patient of Italian nationality who had never been abroad was observed by the Oncohematology and TMO Unit of the Maddalena Clinic for about two months after having a recurrent spiking fever for four days. Regarding the pathological history: erosive gastropathy for which the patient received a transfusion due to anemization (Hb 6.5 mg/dL) in January. Fever started after the transfusion, and a urinary tract infection due to *E. coli* arose, which was treated with ciprofloxacin and subsequently ceftazidime because of the persistence of the febrile state. Physical examination: sensory alertness, fevers up to 38.8 °C accompanied by chills, absence of lymphadenomegaly, and blood pressure (PAO) 90/60. On the first day of hospitalization, the patient underwent microbiological, biochemical–clinical, and instrumental investigations. Regarding the blood: WBC 3390/mmc, Hb 10.5 g/dL, and PLT 131000. Negative abdominal ultrasound. Negative microbiological investigations for toxoplasma, cytomegalovirus (CMV), Epstein–Barr virus (EBV), hepatitis B virus (HBV), hepatitis C virus (HCV), treponema pallidum hemagglutination assay (TPHA), Weil–Felix, and Widal–Wright. Negative urinary culture. An osteomyelobiopsia was performed, which showed hypercellulated bone marrow with reactive type changes. On the third day of hospitalization, the patient has a worsening health condition due to the persistence of fever, thrombocytopenia (reduction of platelets from 131,000 to 48,000), anemia (reduction of hemoglobin from 10.5 to 8.2 mg/dL), and acute renal failure (azotemia 185 mg/dL, creatinine 4.73 mg/dL, uricemia 9.3 mg/dL) for which dialysis was started. Returning to the donor, we learned that he was a missionary priest who had traveled to endemic regions; however, this was more than 10 years ago. Therefore, he had been subjected to the mandatory donation tests according to the “Provisions relating to the quality and safety requirements of blood and blood components,” resulting in a suitable donation [7]. The blood sample was screened for the following markers: HIV antigen antibody, HBsAg, anti-HCV, and syphilis. In relation to the long period since he had been in endemic regions, the donor had not been screened for anti-malarial antibodies. However, based on the donor’s epidemiological history, the patient’s clinical history, and the clinical–laboratory data, blood samples for the antigenic, microscopic, and molecular identification of *Plasmodium* spp. were sent to the Unit of Microbiology and Virology of the Paolo Giaccone Hospital in Palermo. In the peripheral blood smear, trophozoites and schizonts were identified as *P. malariae* via their morphologies. Real-time PCR was used as a support for the microscopic examination and was also used on the donor blood to confirm the causal link. After infectious counseling with the Infectious and Tropical Diseases Unit of the Paolo Giaccone Hospital, therapy began with piperaquine tetraphosphate + dihydroartemisinin (Eurartesim) 320/40, which was followed by an improvement of health with a resolution of fever, recovery of liver and kidney function, and a return of the blood count within normal parameters.

## 3. Discussion

Transfusion-transmitted malaria (TTM) is an accidental *Plasmodium* spp. infection that is caused by a whole blood or blood component transfusion from a malaria-infected donor to a recipient. The first case of malaria as an accidental consequence of blood transfusion was described in 1911 by Woolsey [8]. In Italy, the first reported case dates to 1963 in Liguria and the last case described in Sicily occurred in 2005 in a patient with acute renal failure [9]. The risk of transfusion-transmitted malaria in nonmalaria-endemic countries is principally contributed by blood donors that previously lived or traveled to malaria-endemic countries [10]. *Plasmodium* species were detected in 100 TTM case reports with different frequencies: 45% *P. falciparum*, 30% *P. malariae*, 16% Plasmodium vivax, 4% *P. ovale*, 2% *P. knowlesi*, and 1% mixed infection *P. falciparum/P.malariae* [5]. A significant difference between the natural infection and TTM is that the former undergoes an initial asymptomatic phase (pre-erythrocytic), which allows for the activation of innate immunity cells against malaria parasites that give the naïve host time to develop a more specific protective immunity. Infected blood transfusions directly release malaria parasites in the recipient’s bloodstream such that innate immunity is not activated and the risk of complications increases [11]. Therefore, malaria remains a rare but serious complication of transfusions. The main problem regarding transfusion-transmitted malaria is related to the presence of asymptomatic blood donors that are predominantly “semi-immune” with very low parasitic loads. This suggests that thick and thin blood smears, which are still used today as the gold standard for the diagnosis of malaria, cannot be used for donor screening [12]. These asymptomatic infections may remain undetected [13], and experimental evidence suggests that as few as 10 infected RBCs can be sufficient to transmit the infection [14]. All *Plasmodium* spp. are able to survive in stored blood, even if frozen, and retain their viability for at least 1 week, possibly well over 10 days depending on the conditions of storage [15]. Furthermore, the donors may remain infective for up to 1 year with *P. falciparum*, 3 years with *P. vivax*, and decades with *P. malariae*. [5,6,7,8,9,10,11,12,13,14,15,16] In fact, in the clinical case presented, the patient was infected with *P. malariae* due to a blood transfusion that was received from an asymptomatic donor who had been in endemic regions a decade earlier. Infection was confirmed in the donor using molecular biology. This suggests that all blood donors who were in endemic areas should be screened for anti-malarial antibodies, even if they were there a long time beforehand. To reduce cases of transfusion-transmitted malaria, diverse surveillance strategies can be used, including pre-donation questionnaires and/or laboratory screening [17]. The use of serological and molecular tests could represent important tools in the prevention of diseases [18] but the sensitivity and specificity of screening tests in blood donors remain the object of study. In Italy, the relevant regulation (no. 219, October 2005) currently requires individuals that are considered at risk based on a donor questionnaire to be tested for anti-*Plasmodium* spp. antibodies and, in case of a positive result, to be excluded as blood donors for 3 years [7]. The limits of the serological tests used are related to sensitivity in accordance with a study conducted in 2018 in which five ELISA commercial kits were evaluated that were highly specific (100%), but with a sensitivity between 53 and 64% [19]. Furthermore, serological tests are indirect tests; therefore, they do not necessarily indicate parasitemia and could lead to the exclusion of uninfected donors [20,21]. An investigation by the American Red Cross, the New York State Department of Health, and the Centers for Disease Control and Prevention suggested the importance of applying sensitive laboratory techniques to identify infected donors [3,22,23].

The WHO recommends that nucleic acid amplification tests should be considered for epidemiological research and survey mapping of sub-microscopic infections. Indeed, the polymerase chain reaction (PCR) is the most sensitive method available, detecting parasitemia from 2–5 parasites/μL, unlike microscopy and RDTs, which have a sensitivity of 50–500 parasites/μL. and ~100 parasites/μL respectively. However, PCR is an expensive and complex method [24,25].

## 4. Conclusions

Transfusion-trasmitted malaria is a rare but serious possibility. The optimal strategy to reduce the risk of transfusion-transmitted malaria in non-endemic countries without unnecessary exclusion of blood donations is still debated and semi-immune individuals represent the biggest challenge for TTM screening, as they might become asymptomatic carriers with a very low parasite density, which is difficult to detect with current direct diagnostic methods. Currently, microscopy represents the gold standard analysis method for the diagnosis of malaria despite the limited sensitivity when detecting asymptomatic infections with low parasitemia. Molecular tests and serological tests, combined with a donor’s questionnaire, should be used to prevent transfusion-transmitted malaria in non-endemic areas.

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
