# Peer review of "Transfusion-Transmitted Malaria of Plasmodium malariae in Palermo, Sicily"

_healthcare, 2021, doi:10.3390/healthcare9111558_

Round 1

Reviewer 1 Report

The manuscript presented for review is a case report about the transfusion-transmitted malaria of Plasmodium malariae in Palermo, Sicily. Since malaria transmission through transfusions is extremely rare in non-endemic areas, I find the case report interesting. The course of treatment is described in clearly and detail way. However, the discussion of presented case with other TTM cases should be better conducted (some information on donors, what their diagnosis looked like, etc.). Although the subject of the work is interesting, the manuscript has some errors which are listed below.

Line 87: Plasmodium spp à Plasmodium spp.

Line 96: An important difference à A significant differences

Line 99: gines à give

Line 121: In Italy à In Italy,

Author Response

Dear reviewer,

I would like to thank you for the opportunity of editing the manuscript for Healthcare.

My co-authors and I apologize being previously unclear and we hope that the changes we made meet the yours demands.

In what follows we provide the list of the changes we made (responses are marked as “R” while the changes are interline in green in the text)

The manuscript presented for review is a case report about the transfusion-transmitted malaria of Plasmodium malariae in Palermo, Sicily. Since malaria transmission through transfusions is extremely rare in non-endemic areas, I find the case report interesting. The course of treatment is described in clearly and detail way. However, the discussion of presented case with other TTM cases should be better conducted (some information on donors, what their diagnosis looked like, etc.). Although the subject of the work is interesting, the manuscript has some errors which are listed below.

Line 87: Plasmodium spp à Plasmodium spp.

  1. Done

Line 96: An important difference à A significant differences

  1. Done

Line 99: gines à give

  1. Done

Line 121: In Italy à In Italy,

  1. Done

Reviewer 2 Report

The manuscript is well arranged and is important in relation to transfusion-transmitted malaria. I consider it appropriate for this journal.

However, there are a couple of suggestions that might be addressed.

Line 72. Please include which mandatory donation tests were performed. The World Health Organization (WHO) recommends that all blood donations should be screened for malaria; there is no consensus about which test should be used.

One important piece of information that is missing is whether the donor had malaria ten years ago. If yes, by which Plasmodium species. Normally, donors who have had P. malariae are permanently excluded.

Lines 121-124. Are the recommendations from European guidelines used in Italy? (according to http://www.ema.europa.eu/docs/en_GB/document_library/Regulatory_and_procedural_guideline/2009/10/WC500004484.pdf.). Is the regulation no. 219 different? Please insert a brief paragraph with this issue and all the aspects of the recommendations (about malaria) that are in course in the country. As mentioned above, normally donors who have had P. malariae are permanently excluded.

There are many scientific names (Plasmodium or Plasmodium species) without italic. Please check the entire document.

Line 108. ….experimental evidence suggests that as few as 10 infected RBCs can be sufficient to transmit the infection”. Please insert the reference. The same after storage (line 110).

Line 136. However, it is an expensive and complex method. [22-23]. Replace “it” by “PCR”.

There are dots (full stops) before the references. Please, check all the manuscript and remove them.

Line 144. Remove the comma after microscopy.

Finally, after suspicion, why was serological testing not performed in parallel with the antigenic, microscopic and molecular tests?  It would be interesting to know if the donor had anti-Plasmodium antibodies and if performing a serological test would have prevented TTM.

Author Response

Dear reviewer,

I would like to thank you for the opportunity of editing the manuscript for Healthcare.

My co-authors and I apologize being previously unclear and we hope that the changes we made meet the yours demands.

In what follows we provide the list of the changes we made (responses are marked as “R” while the changes are interline in green in the text)

The manuscript is well arranged and is important in relation to transfusion-transmitted malaria. I consider it appropriate for this journal.

However, there are a couple of suggestions that might be addressed.

Line 72. Please include which mandatory donation tests were performed. The World Health Organization (WHO) recommends that all blood donations should be screened for malaria; there is no consensus about which test should be used.

  1. The donor during the anamnestic survery had omitted that he had had malaria about 10 years ago. In Italy the blood samples are screened for malaria only if the donor report a past infection according to italian guide line.

The sentence with mandator tests performed was added.

One important piece of information that is missing is whether the donor had malaria ten years ago. If yes, by which Plasmodium species. Normally, donors who have had P. malariae are permanently excluded.

  1. The donor had acquired the infection during a mission in India, but he did not remember the Plasmodium species

Lines 121-124. Are the recommendations from European guidelines used in Italy? (according to http://www.ema.europa.eu/docs/en_GB/document_library/Regulatory_and_procedural_guideline/2009/10/WC500004484.pdf.). Is the regulation no. 219 different? Please insert a brief paragraph with this issue and all the aspects of the recommendations (about malaria) that are in course in the country. As mentioned above, normally donors who have had P. malariae are permanently excluded.

  1. In Italy we used the guidelines of the Italian ministry of health. The guidelines are reported in the following link: https://www.gazzettaufficiale.it/eli/id/2015/12/28/15A09709/sg

There are many scientific names (Plasmodium or Plasmodium species) without italic. Please check the entire document.

  1. Done

Line 108. ….experimental evidence suggests that as few as 10 infected RBCs can be sufficient to transmit the infection”. Please insert the reference. The same after storage (line 110).

  1. Done. The references were added.

Line 136. However, it is an expensive and complex method. [22-23]. Replace “it” by “PCR”.

  1. Done

There are dots (full stops) before the references. Please, check all the manuscript and remove them.

  1. Done

Line 144. Remove the comma after microscopy.

  1. Done

Finally, after suspicion, why was serological testing not performed in parallel with the antigenic, microscopic and molecular tests?  It would be interesting to know if the donor had anti-Plasmodium antibodies and if performing a serological test would have prevented TTM.

  1. Serological test performed on serum of donor, after the case and after a second anamnestic survey, has relevated the anti-Plasmodium antibodies